# Longitudinal Association between Stressful Life Events and Suicidal Ideation in Adults with Major Depression Disorder: The Mediating Effects of Insomnia Symptoms

**DOI:** 10.3390/bs14060467

**Published:** 2024-05-31

**Authors:** Ya Chen, Xue Han, Yingchen Jiang, Yunbin Jiang, Xinyu Huang, Wanxin Wang, Lan Guo, Ruirui Xia, Yuhua Liao, Huimin Zhang, Kayla M. Teopiz, Roger S. McIntyre, Beifang Fan, Ciyong Lu

**Affiliations:** 1Department of Medical Statistics and Epidemiology, School of Public Health, Sun Yat-sen University, Guangzhou 510080, China; cheny2253@mail2.sysu.edu.cn (Y.C.);; 2Guangdong Provincial Key Laboratory of Food, Nutrition and Health, Sun Yat-sen University, Guangzhou 510080, China; 3Guangdong Engineering Technology Research Center of Nutrition Translation, Guangzhou 510080, China; 4Department of Psychiatry, Shenzhen Nanshan Center for Chronic Disease Control, Shenzhen 518054, China; xuehan_sz@hotmail.com (X.H.);; 5Brain and Cognition Discovery Foundation, Toronto, ON M2J 4A6, Canada; 6Department of Pharmacology, University of Toronto, Toronto, ON M2J 4A6, Canada; 7Department of Psychiatry, University of Toronto, Toronto, ON M2J 4A6, Canada

**Keywords:** stressful life events, insomnia, suicidality, mortality, major depression disorder

## Abstract

Stressful life events (SLEs) and suicidal ideation (SI) are prevalent in persons with major depression disorder (MDD). Less is known about the underlying role of insomnia symptoms in the association between SLEs and SI. This three-wave prospective cohort study sought to investigate the longitudinal association among SLEs, insomnia symptoms, and SI in persons with MDD. The study population included 511 persons with MDD (mean [SD] age, 28.7 [6.7] years; 67.1% were females). Generalized estimated equations (GEEs) were utilized to explore prospective association among exposure of SLEs, insomnia symptoms, and SI. Additionally, a structural equation model (SEM) was employed to estimate the longitudinal mediating effect of insomnia symptoms in the relationship between SLEs and SI. Our study demonstrated that cumulative SLEs were determined to be longitudinally associated with SI in persons with MDD. We further observed that the association between SLEs and SI was significantly mediated by insomnia symptoms. Clinicians assessing persons with MDD, especially those with the history of SLE, could carefully evaluate and promptly treat insomnia symptoms as part of personalized assessment of their depressive illness, thereby achieving early prevention and intervention for suicidal behaviors in persons with MDD.

## 1. Introduction

Major depressive disorder (MDD) is an increasingly serious public health issue [1] and is associated with suicidal behavior [2]. An estimated 90% of persons who die by suicide suffer from one or more mental diseases, with MDD accounting for 59–87% of all reported suicides [2]. Suicidal ideation (SI) is the main precondition of suicidal act [3] and represents an elevated risk of future suicide [4]. A recent meta-analysis of 53,598 persons with MDD reported that the overall prevalence of SI was 37.7% [2] and the rate is increasing [5]. Furthermore, one-fifth of persons with MDD and SI developed a future suicide attempt [6]. Therefore, identifying persons with MDD who are at high risk of SI early has important clinical implications [7].

The integrated motivational–volitional (IMV) model of suicidality describes that individual vulnerabilities confer increased risk for developing SI when activated by the presence of stressors [8]. Stressful life events (SLEs) are common stressors for suicide [9]. SLEs refer to significant incidents that occur abruptly and provoke strong psychological responses in individuals [10]. However, the current evidence on the association between SLEs and SI requires larger epidemiological confirmation.

First, conclusions about the cumulative number of SLEs or the effect of high perceived stress of SLEs on SI were inconsistent. Longitudinal studies in adults with depression have found that it was individuals’ high perception of stress rather than the number of SLEs that increased the risk of suicidal behavior [11]. Longitudinal studies in college students have found an association between high perceived stress and SI [12]. However, Wang et al. also reported positive associations between the number of SLEs and SI [13].

Second, prior studies have investigated risk factors associated with suicidal behaviors in different populations, including specific stressors such as childhood trauma [14,15,16], interpersonal stress [17,18], bullying [19], violence [20], and so on. However, there is limited research examining and comparing different types and periods of stressors comprehensively within the same study, which makes it difficult to identify what life events affect persons with MDD the most.

Trauma-induced insomnia may be a precursor to other disabling post-traumatic complications. The stress-psychopathological model suggests that SLEs are associated with suicidal behavior, as well as psychopathological factors that may also influence the association between SLEs and SI [21]. For example, SLEs may be predisposing factors for insomnia which may bridge the association between SLEs and SI [22]. Short et al. also reported that adults who have experienced trauma were twice as likely to develop insomnia as adults who have not been exposed to trauma [23]. Furthermore, people with persistent insomnia had higher rates of SI [24,25]. Evidence from cross-sectional studies in adolescents and individuals with bipolar disorder has indicated that insomnia symptoms mediated the association between SLEs and SI [26,27]. However, longitudinal studies exploring the potential role of insomnia symptoms between SLEs and SI in persons with MDD are still lacking.

Given that insomnia may be a potentially modifiable risk factor for SI, a better understanding of SLEs and the effects of insomnia symptoms on SI may be helpful for early intervention of suicidal behaviors. Therefore, our aims were to examine (1) the effect of the cumulative number and types of SLEs on SI in persons with MDD and (2) the longitudinal, mediational role of insomnia symptoms in the association between SLEs and SI in persons with MDD.

## 2. Materials and Methods

### 2.1. Sample and Procedure

Data were collected from a subgroup cohort (patient cohort) of Depression Cohort in China (DCC, ChiCTR registry number: 1900022145), which is a large, ongoing cohort study targeting persons aged 18–64 years with MDD and the design of DCC study has been elaborated in a previous study [28]. In the primary study, participants aged 18–64 years were enrolled from a mental health specialty hospital (Shenzhen Kangning Hospital) between June 2020 and September 2022. The inclusion criteria of this study were as follows: (1) diagnosed with major depressive disorder (MDD) confirmed by certified psychiatrists using the Mini-International Neuropsychiatric Interview (M.I.N.I.) and the Diagnostic and Statistical Manual of Mental Disorders 5th edition (DSM-5); (2) score on the 17-item Hamilton Depression Rating Scale (HAMD-17) ≥ 8 and the Patient Health Questionnaire-9 (PHQ-9) ≥ 5; (3) being capable of understanding research questionnaires and providing informed consent for themselves. The study protocol was approved by the institutional review board of Sun Yat-sen University School of Public Health (Ethical code: L2017044), and written informed consent was acquired from all individuals before participating in the study.

The study flow is depicted in Figure 1. At baseline, participants completed self-reported questionnaires concerning sociodemographic information, health related behaviors, mental health status, SLEs, SI, and insomnia symptoms. Participants were followed up at 12 and 24 weeks and the information about insomnia symptoms and SI was collected. Of the 1097 participants who completed the baseline survey, 118 participants dropped out of the study, 33 participants were diagnosed with bipolar disorder, 198 participants could not be contacted or have incomplete questionnaires, and 237 participants only attended the 12-week or 24-week follow-up, resulting in 511 participants being included in the final analysis.

### 2.2. Measures

#### 2.2.1. Stressful Life Events

The participants’ previous exposure to SLEs was assessed using the stressful life events screening questionnaire (SLESQ) [29], which has demonstrated good reliability and validity [30]. The SLESQ consisted of 13 items, which asked participants to report SLEs that occurred in their daily life. Given the national context of China, following discussions with pertinent experts, we excluded the item “Living in environments akin to armed conflict or war zones where life-threatening dangers could occur at any moment.” Consequently, this decision led to the development of a 12-item Chinese version of the SLESQ, which was utilized in this study to assess exposure to SLEs [31]. A “yes” answer to any of the 12 items was recorded as positive, meaning that they had experienced SLEs. The response to each type of SLE determined whether it occurred and was recoded (0 = did not occur, 1 = occurred), and the cumulative number of SLEs will be accumulated (the total scores ranging from 0–12). Furthermore, due to the significant skewness observed in the measurement of SLEs, with the majority (>75%) of participants experiencing 0 to 3 types of SLEs, we defined 3 levels of SLE exposure: 0 SLE as low exposure, 1 or 2 SLEs as normative exposure, and 3 or more SLEs as high exposure. This classification aimed to ensure a relatively even distribution across groups [32]. Additionally, the SLESQ was utilized at baseline and showed acceptable internal consistency in the current study (Cronbach’s *α* = 0.70).

#### 2.2.2. Suicidal Ideation

The severity of SI in the latest week was measured by the Beck Scale for Suicide Ideation (BSSI) [33]. This self-report instrument includes 19 items that are scored based on a 3-point Likert scale (score range, 0–2, 0 = “I have no wish to die”; 1 = “I have a weak wish to die”; 2 = “I have a moderate-to-strong wish to die”). The intensity of SI is assessed by the first 5 items. The risk of suicide is evaluated based on items 6 to 19, which determine the likelihood of a participant with SI actually attempting suicide. Items 6–19 are only completed if the respondent rates items 4 or 5 with a score of 1 or greater, and thus, we had more complete data for the first 5 items. Consequently, in this study, SI was assessed by the first 5 items and higher scores representing higher levels of SI. Participants was considered to be free of SI only when they received a score of 0 for both items 4 and 5. The BSSI was utilized at baseline and at 12 and 24 weeks, and showed good internal consistency in the current study (Cronbach’s α = 0.81 at baseline; Cronbach’s α = 0.85 at 12 weeks; Cronbach’s α = 0.83 at 24 weeks).

#### 2.2.3. Insomnia Symptoms

The Insomnia Severity Index (ISI) was utilized to evaluate insomnia symptoms and has shown good reliability and validity in Chinese populations [34]. It consists of 7 measurements items that assess difficulty with initiating or maintaining sleep, degree of early awakening, satisfaction with sleep, impact of sleep difficulties on daily routines, distress, and noticeability of sleep disorder to others over the past 2 weeks [35]. The scale employs a 5-point Likert scale, ranging from 0 (no problem) to 4 (extremely severe problem). The total scores range from 0 to 28, where higher scores indicate more severe insomnia symptoms. Insomnia symptoms were assessed at baseline and at 12 and 24 weeks and showed good internal consistency in the current study (Cronbach’s α = 0.90 at baseline; Cronbach’s α = 0.93 at 12 weeks; Cronbach’s α = 0.93 at 24 weeks).

#### 2.2.4. Covariates

Factors associated with SLEs or SI were taken into account among persons with MDD. The covariates consist of sociodemographic information, health related behaviors, and mental health status.

Sociodemographic information included gender, age, marriage, education level, employed status, and family monthly income. Health-related behaviors included lifetime smoking, lifetime drinking, and weekly exercise habit. Lifetime smoking, life drinking, and weekly exercise habit were evaluated by the following questions: “Have you ever smoked a cigarette (0 = No; 1 = Yes)? Have you ever consumed at least one alcoholic drink of any kind (0 = No; 1 = Yes)? Do you have a weekly exercise for more than 30 min (0 = No; 1 = Yes)?”. Mental health status included depressive symptoms, anxiety symptoms, and resilience. Depression symptoms and anxiety symptoms were evaluated using the Patient Health Questionnaire-9 (PHQ-9, Cronbach’s alpha  =  0.81 in this study) and the Generalized Anxiety Disorder Scale-7 (GAD-7, Cronbach’s alpha  =  0.90 in this study), respectively, which have been validated and widely administered in Chinese populations with reliable psychometric properties [36,37]. We dichotomized GAD-7 and PHQ-9 scores into measures of moderate or severe anxiety and depression symptoms (total scores ≥10) [38,39]. Resilience was measured with the Connor–Davidson Resilience Scale (CD-RISC, Cronbach’s α = 0.92 in this study). Considering the cut-point score in the Chinese population [40], a total score of above 60 points indicates good resilience.

### 2.3. Statistical Analysis

First, descriptive analyses stratified by SLEs were used to describe the sample characteristics, and the data were presented as number (%) and mean (standard deviation, SD). The *t* tests for continuous variables and the Chi-square tests for categorical variables were performed to compare the differences between the two groups (exposed to and not exposed to SLEs).

Second, generalized estimated equations (GEEs) were employed to explore the longitudinal association of different types and numbers of SLEs with SI before and after adjusting for confounding variables. Factors that were significant in univariate analyses were considered as confounding variables. The results were expressed as adjusted *β* estimates and 95% confidence intervals. A linear link function for the continuous outcome variable and an unstructured correlation working structure were used in above analysis.

Third, structural equation models (SEMs) using the maximum likelihood (ML) method were employed to test the mediating effects of insomnia symptom on the association between SLEs and SI after adjusting for covariates and SI at baseline. As the variables relating to SLEs, insomnia symptoms, and SI were continuous, the standardized path coefficients, standardized total effects, and standardized indirect effects were reported and the bias-corrected 95% confidence intervals (CIs) were obtained with 1000 bootstraps resamples. All statistical tests were two-tailed, and statistical significance was set at *p* < 0.05. Statistical analyses used IBM SPSS Statistics (V.27, IBM Corporation, New York, NY, USA) and Mplus 8.3 (Muthén and Muthén, Los Angeles, CA, USA).

## 3. Results

### 3.1. Sample Characteristics

Table 1 shows the sample characteristics of this study. Among the 511 participants, 66.9% reported experiencing SLEs. The mean (SD) age of the participants was 28.7 (6.7) years, with 67.1% being females. Additionally, 74.2% of the participants were unmarried, 56.9% had an education level of undergraduate or above, 68.9% were employed, and 56.0% reported a household monthly income above CNY 10,000. Compared with the participants who were not exposed to SLEs, the group exposed to SLEs had a higher proportion of smoking behaviors (*p* < 0.05), more severe insomnia symptoms (*p* < 0.05), and higher SI scores (*p* < 0.01). Except for age, gender, and occupation, there were no significant differences in baseline characteristics between eligible participants and ineligible participants (Appendix A).

The proportion of SLEs is presented in Appendix A. According to the participants’ response on the SLESQ, the proportion of SLEs was higher in being humiliated or discriminated (45.4%), extreme fear or helplessness (42.1%), and childhood physical abuse (18.6) compared to other types of SLE.

### 3.2. Longitudinal Association of Types of SLEs with SI

As shown in Table 2, the univariate analysis showed that without adjusting for other variables, three or more events and insomnia symptoms were positively associated with SI (*p* < 0.01). After adjusting for control variables and insomnia symptoms, the multivariable analysis indicated that only three or more events were significantly associated with an elevated risk of SI (*p* < 0.05).

### 3.3. Longitudinal Association of Types of SLEs with SI

As shown in Table 3, the univariate analysis revealed that bereavement, witnessing a traumatic event, childhood physical abuse, adulthood physical abuse, being threatened, humiliated, or discriminated, and extreme fear or helplessness were all associated with an increased risk of SI (all *p*  <  0.05). After controlling for potential confounding variables, the multivariate analysis indicated that only childhood physical abuse was positively associated with an increased risk of SI (adjusted *β* estimate = 0.26, 95% CI = 0.02–0.51).

### 3.4. Mediating Effects of Insomnia Symptoms

Structural equation models were performed with the total numbers of SLEs at baseline as an independent variable, SI at 24 weeks as a dependent variable, and insomnia symptoms at 12 weeks as a mediating factor. As presented in Table 4, the standardized direct path coefficients of SLEs on SI (standardized *β* estimate = 0.100, 95% CI = 0.010–0.193) were statistically significant (*p* < 0.05).

The standardized indirect effects of SLEs on SI (standardized *β* estimate = 0.026, 95% CI = 0.003–0.053) mediated by insomnia symptoms were also statistically significant (*p* < 0.05) and the mediating effect accounted for 20.6%, indicating that insomnia symptoms partially mediated the associations between SLEs and SI. The results of SEM models depicting the mediating roles of insomnia symptoms in the association between SLEs and SI are illustrated in Figure 2.

Moreover, the mediating role of insomnia symptoms in the association between specific types of SLEs and SI was examined. However, no significant mediating effect of insomnia symptoms was found in any specific type of SLEs (Appendix A).

## 4. Discussion

To our knowledge, this is one of the few longitudinal studies that examines the relationship among SLEs, insomnia symptoms, and SI in persons with MDD. The findings of our study indicate that the larger the cumulative number of SLEs experienced by persons with MDD, the greater the risk of SI. Furthermore, insomnia symptoms may play a longitudinal mediating role in the association between SLEs and SI.

Suicide has been conceptualized as a response to life stressors. Our results suggest that the proportion of persons with MDD exposed to SLEs was relatively high (66.9%), which was in accordance with a separate study conducted in persons with MDD in China (65.6%) [41], but lower than the study conducted in US veterans with MDD (83.0%) [42] as well as a separate study conducted in persons with MDD in Sweden (83.1%) [43].

In addition, as illustrated in Appendix A, SI was also common among persons with MDD, with the prevalence of 79.3% in our study. A recent study reported that 66.0% of participants with MDD had SI [7], which was lower than in our study. Methodological differences could account for foregoing discrepancies, including sample composition and various assessment tools of SLEs and SI. Cultural and racial differences in the disclosure of SLEs and SI may also contribute to the differences observed [44]. Overall, considering the high prevalence of both SLEs and SI in persons with MDD, it is crucial to prevent the risk of serious life-threatening outcomes that may be associated with SLEs.

After controlling for demographic characteristics and other psychosocial covariates, the risk of SI increased with the cumulative numbers of SLEs experienced. Our results were consistent with the extant literature that has documented the association between cumulative stressors and SI. For example, cross-sectional studies in people with bipolar disorder have reported that as the number of SLEs experienced over a lifetime increased, affected individuals exhibit more severe psychopathological symptoms, such as more severe anxiety symptoms, higher rates of suicidal tendencies, and greater interpersonal relationship problems [45].

Additionally, a prospective study conducted by Wang et al. found a dose–response relationship between recent SLEs and SI in persons with MDD [13]. A growing body of research has shown that the cumulative effect of psychosocial stressors across the lifespan could increase the likelihood of developing mental health problems [46]. Moreover, it has been reported that the effect of perceived stress of stressors has an independent effect on an individual’s suicide risk, suggesting that the impact of SLEs on suicide risk may be influenced by an individual’s cognitive assessment of life events [11,47,48].

The aforementioned studies emphasize individual variations, suggesting that different individuals respond differently when faced with the same stressor, and these responses can have a significant impact on the problem-solving level of appraisal following an SLE. Furthermore, a systematic review also reported that the magnitude of the effect of multiple SLEs or an individual SLE in a given study is influenced by the type of SLE screening framework [49].

Overall, there are mixed results reported on the effect of the number of SLEs on SI. Our results add and extend the extant literature documenting the longitudinal association between the cumulative number of SLEs and SI. Future studies could explore the effects of the numbers of life events and high perceived stress on SI simultaneously, as well as comparative studies on the effects of single or multiple stressors on SI using validated screening tools.

We also examined how specific stressors across the lifespan may influence the risk of SI. After assessing both childhood and adult stressors, our study reported that traumatic experiences, including childhood physical maltreatment, were particularly associated with the risk of SI. Life adversities during critical developmental stages of life, especially childhood, are associated with more severe and persistent symptoms that negatively affect health outcomes [50]. In addition, it is reported that physical abuse could increase the risk of suicide tendency in adolescents [51], young people [52], persons with MDD [53], bipolar disorder [54], midlife [55], and veterans [42], indicating that childhood physical trauma experiences have a strong impact on suicide risk. The results from our study are in accordance with prior studies, wherein a history of exposure to childhood physical abuse was significantly associated with suicidal behaviors.

Furthermore, our study indicated that SLEs are associated with insomnia symptoms, and may further mediate SI. The results of our study align with the findings of previous cross-sectional studies conducted. For example, Yang et al. reported a negative correlation between childhood maltreatment, sleep duration, and SI among Chinese young adults [56]. A separate cross-sectional study involving 11,831 adolescents indicated that the association between SLEs and suicidality was partially mediated by insomnia [26]. Moreover, a similar association was observed in persons with mental disorders. A study evaluating adolescents with depression has also reported that insomnia is a mediator of childhood physical neglect leading to SI [57]. Palagini et al. (2020) also reported that insomnia symptoms may contribute to the association between early life stressors and SI in adults with bipolar disorder [58]. Moreover, a recent three-year prospective study including 6995 adolescents found that a partial mediation of insomnia 1 year later in the relationship between baseline life stress and SI 2 years later [59], which in accordance with our results herein.

There is yet no clear understanding of the mechanisms subserving the association between SLEs and SI in persons with MDD. One possible mechanism could be the dysregulation of the hypothalamic–pituitary–adrenal (HPA) axis. After exposure to stressors, the activation of the HPA axis triggers an excessive release of cortisol [60], which may induce increased sleep reactivity and excessive wakefulness in susceptible individuals, leading to insomnia symptoms such as difficulty falling asleep and shortened sleep time [61].

Studies conducted in persons with insomnia have also reported that high 24-h urinary cortisol levels were associated with increased total wake time in patients with chronic insomnia. For example, people with insomnia symptoms who slept less than 5 h had higher morning cortisol levels than those who slept more than 5 h [62]. It is also separately reported that elevated cortisol levels could increase the risk of suicide [63]. Future research is needed to demonstrate this possible mechanism.

This study’s findings should be interpreted with several limitations. First, the collection of data was performed through a self-report questionnaire, which may be vulnerable to recall bias. Second, the current study mainly captured the types and cumulative number of SLEs but paid less attention to the extent to which they affected participants. Future studies could explore whether the numbers and impact extent of SLEs have different effects on SI in person with MDD or not. Third, we mainly focused on the effect of negative stressful life events on SI and did not take positive life events into account. In addition, our study has a relatively small sample size, and the sample was drawn only from the Nanshan District of Shenzhen, thus lacking extrapolation and representation. The sample size could be expanded for further study.

## 5. Conclusions

Herein, our study illustrated that childhood physical abuse and the cumulative number of SLEs were longitudinally associated with SI in persons with MDD. Furthermore, insomnia symptoms played a longitudinal mediating role on the relationship between SLEs and SI. Prevention, screening, and early intervention strategies for insomnia symptoms in persons with MDD who experience SLEs may prevent SI, a more serious mental health outcome.

## Figures and Tables

**Figure 1 behavsci-14-00467-f001:**
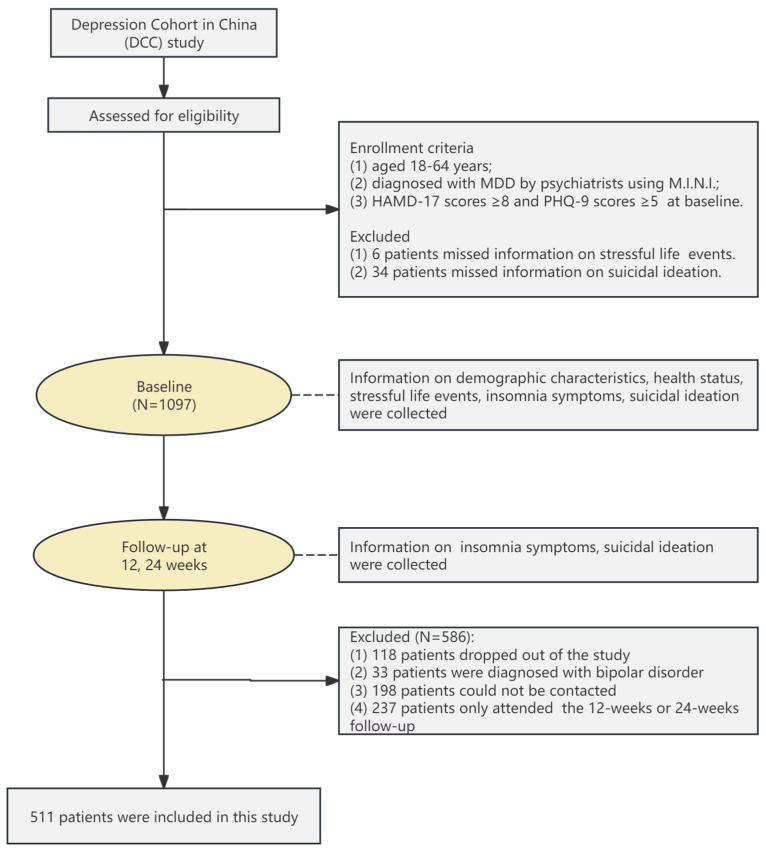
Participant recruitment and flow.

**Figure 2 behavsci-14-00467-f002:**
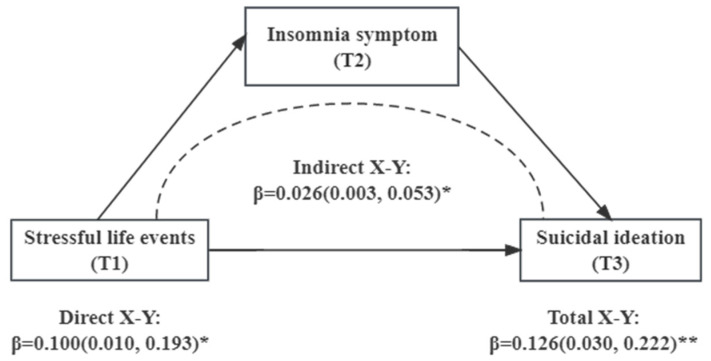
Mediating effects of insomnia symptoms on the relationship between SLEs and SI. Note: *: *p* <0.05; **: *p* <0.01.

**Table 1 behavsci-14-00467-t001:** Baseline characteristics of 511 participants.

Factor	Total, *n* (%)	Stressful Life Events		*ꭓ* ^2^	*p* Value
		Yes	No		
Total	511 (100.0)	342 (66.9)	169 (33.1)	NA	NA
Gender (*n*, %)					
Male	168 (32.9)	117 (34.2)	51 (30.2)	0.834	0.361
Female	343 (67.1)	225 (65.8)	118 (69.8)		
Age (mean, SD)	28.7 (6.7)	28.6 (6.1)	28.8 (7.0)	0.228	0.820
Marriage (*n*, %)					
Unmarried	379 (74.2)	261 (76.3)	118 (69.8)	2.808	0.094
Married	122 (23.9)	74 (21.6)	48 (28.4)		
Missing	10 (2.0)	7 (2.0)	3 (1.8)		
Education level (*n*, %)					
Below undergraduate	220 (43.1)	155 (45.3)	65 (38.5)	2.171	0.141
Undergraduate or above	291 (56.9)	187 (54.7)	104 (61.5)		
Employed status (*n*, %)					
Unemployed	158 (30.9)	109 (31.9)	49 (29.0)	0.466	0.495
Employed	352 (68.9)	232 (67.8)	120 (71.0)		
Missing	1 (0.2)	1 (0.3)	0 (0.0)		
Family monthly income (*n*, %)					
<CNY 10,000	187 (36.6)	132 (38.6)	55 (32.5)	0.756	0.385
≥CNY 10,000	286 (56.0)	191 (55.8)	95 (56.2)		
Missing	38 (7.4)	19 (5.6)	19 (11.2)		
Smoking (*n*, %)					
No	301 (58.9)	189 (55.3)	112 (66.3)	5.169	0.023
Yes	207 (40.5)	150 (43.9)	57 (33.7)		
Missing	3 (0.6)	3 (0.9)	0 (0.0)		
Drinking (*n*, %)					
No	79 (15.5)	49 (14.3)	30 (17.8)	0.987	0.320
Yes	431 (84.3)	292 (85.4)	139 (82.2)		
Missing	1 (0.2)	1 (0.3)	0 (0.0)		
Exercise habit per week (*n*, %)					
No	329 (64.4)	224 (65.5)	105 (62.1)	0.625	0.429
Yes	181 (35.4)	117 (34.2)	64 (37.9)		
Missing	1 (0.2)	1 (0.3)	0 (0.0)		
Moderate or severe depressive symptoms (*n*, %)					
No	29 (5.7)	14 (4.1)	15 (8.9)	4.832	0.028
Yes	482 (94.3)	328 (95.9)	154 (91.1)		
Moderate or severe anxiety symptoms (*n*, %)					
No	113 (22.1)	72 (21.1)	41 (24.3)	0.648	0.421
Yes	397 (77.7)	269 (78.7)	128 (75.7)		
Missing	1 (0.2)	1 (0.3)	0 (0.0)		
Resilience at baseline (*n*, %)					
Weak resilience (≤60)	471 (92.2)	317 (92.7)	154 (91.1)	0.097	0.755
Greater resilience	34 (6.7)	22 (6.4)	12 (7.1)		
Missing	6 (1.2)	3 (0.9)	3 (1.8)		
SI at baseline (mean, SD)	3.8 (2.6)	4.0 (2.7)	3.4 (2.5)	−2.703	0.007
ISI scores (mean, SD)	14.9 (6.3)	15.0 (6.6)	14.8 (6.2)	−0.278	0.781

Note: 95% CI = 95% confidence intervals; SI = suicidal ideation; ISI = Insomnia Severity Index.

**Table 2 behavsci-14-00467-t002:** Association between the numbers of SLEs, insomnia symptoms and SI among persons with MDD.

Factor	Suicidal Ideation			
	Model 1		Model 2	
	*β* Estimate (95% CI)	*p* Value	Adjusted *β* Estimate (95% CI)	*p* Value
Number of SLEs at baseline				
0	0 [Reference]		0 [Reference]	
1–2	0.17 (−0.29, 0.62)	0.467	0.19 (−0.01, 0.39)	0.070
≥3	1.37 (0.82, 1.92)	<0.001	0.26 (0.02, 0.51)	0.037
ISI scores (1 score increased)	0.16 (0.13, 0.18)	<0.001	0.08 (0.07, 0.10)	<0.001
Gender				
Male	0 [Reference]		0 [Reference]	
Female	0.31 (−0.13, 0.76)	0.158	0.04 (−0.17, 0.25)	0.721
Age, y	−0.09 (−0.13, −0.06)	<0.001	−0.01 (−0.03, 0.00)	0.084
Marriage				
Unmarried	0 [Reference]		0 [Reference]	
Married	−0.80 (−1.24, −0.39)	<0.001	0.04 (−0.21, 0.28)	0.783
Education level				
Below undergraduate	0 [Reference]		0 [Reference]	
Undergraduate or above	−0.94 (−1.35, −0.51)	<0.001	−0.14 (−0.34, 0.06)	0.158
Employed status				
Unemployed	0 [Reference]		0 [Reference]	
Employed	−0.82 (−1.27, −0.36)	0.001	0.06 (−0.17, 0.29)	0.622
Family monthly income				
<CNY 10,000	0 [Reference]		0 [Reference]	
≥CNY 10,000	−0.97 (−1.39, −0.54)	<0.001	−0.11 (−0.30, 0.10)	0.325
Smoking				
No	0 [Reference]		0 [Reference]	
Yes	0.92 (0.52, 1.32)	<0.001	−0.04 (−0.26, 0.17)	0.676
Drinking				
No	0 [Reference]		0 [Reference]	
Yes	0.95 (0.60, 1.30)	<0.001	0.33 (0.08, 0.58)	0.010
Exercise habit per week				
No	0 [Reference]		0 [Reference]	
Yes	−0.92 (−1.24, −0.60)	<0.001	−0.20 (−0.40, 0.00)	0.053
Moderate or severe depressive symptoms (PHQ-9 ≥ 10) at baseline				
No	0 [Reference]		0 [Reference]	
Yes	1.49 (0.90, 2.09)	<0.001	−0.02 (−0.33, 0.28)	0.868
Moderate or severe anxiety symptoms (GAD-7 ≥ 10) at baseline				
No	0 [Reference]		0 [Reference]	
Yes	0.68 (0.23, 1.13)	0.003	−0.19 (−0.40, 0.04)	0.106
Resilience at baseline				
Weak resilience	0 [Reference]		0 [Reference]	
Greater resilience	−1.71 (−2.30, −1.11)	<0.001	−0.20 (−0.53, 0.13)	0.235

Note: SLEs = stressful life events; 95% CI = 95% confidence intervals; SI = suicidal ideation; ISI = Insomnia Severity Index. Model 1 was the unadjusted association between factor and SI; Model 2 was adjusted for gender, age, marriage, education level, employed status, family monthly income, smoking, drinking, exercise habit, depressive symptom, anxiety symptom, resilience, SI at baseline, and insomnia symptoms at all time points.

**Table 3 behavsci-14-00467-t003:** Association between the types of SLEs and SI in unadjusted model and adjusted model.

Types of SLEs	Suicidal Ideation			
	Model 1		Model 2	
	*β* Estimate (95% CI)	*p* Value	Adjusted *β* Estimate (95% CI)	*p* Value
01. Life-threatening disease (yes vs. no)	−0.11 (−1.02, 0.80)	0.818	−0.11 (−0.43, 0.22)	0.518
02. Life-threatening accident	0.18 (−0.62, 0.97)	0.658	−0.04 (−0.33, 0.26)	0.810
03. Physical assault	0.38 (−1.05, 1.81)	0.601	0.29 (−0.15, 0.74)	0.198
04. Bereavement	0.69 (0.06, 1.32)	0.033	−0.19 (−0.49, 0.12)	0.241
05. Rape	0.15 (−0.94, 1.24)	0.787	−0.42 (−1.05, 0.22)	0.202
06. Other sexual assault	1.04 (−0.27, 2.35)	0.121	−0.09 (−0.67, 0.49)	0.751
07. Witnessed a traumatic event	1.26 (0.60, 1.91)	<0.001	0.18 (−0.11, 0.47)	0.219
08. Childhood physical abuse	0.94 (0.35, 1.53)	0.002	0.26 (0.02, 0.51)	0.034
09. Adulthood physical abuse	1.87 (0.83, 2.91)	<0.001	0.29 (−0.19, 0.77)	0.238
10. Threatened	1.75 (0.12,3.38)	0.036	0.43 (−0.21, 1.07)	0.191
11. Humiliated or discriminated	0.58 (0.15, 1.00)	0.008	0.11 (−0.06, 0.29)	0.218
12. Extreme fear or helplessness	1.02 (0.60, 1.44)	<0.001	0.16 (−0.03, 0.35)	0.101

Note: 95% CI = 95% confidence intervals; SLEs = stressful life events; SI = suicidal ideation. Model 1 was the unadjusted association between factor and SI; Model 2 was adjusted for gender, age, marriage, education level, employed status, family monthly income, smoking, drinking, exercise habit, depressive symptom, anxiety symptom, resilience, SI at baseline, and insomnia symptoms at all time points.

**Table 4 behavsci-14-00467-t004:** Mediating effects of insomnia symptoms on the relationship between SLEs and SI.

Variables	Suicidal Ideation ^1^	
	Standardized *β* Estimate (95%CI)	*p* Value
**Stressful life events (SLEs)**		
**Path**		
SLEs (T1)→Insomnia symptoms (T2)	0.109 (0.010, 0.207)	0.030
Insomnia symptoms (T2)→Suicidal ideation (T3)	0.236 (0.156, 0.309)	<0.001
SLEs (T1)→Suicidal ideation (T3)	0.100 (0.010, 0.193)	0.031
**Effects**		
Indirect effects	0.026 (0.003, 0.053)	0.043
Total effects	0.126 (0.030, 0.222)	0.009
Proportion mediated, %	20.6	

Note: 95% CI = 95% confidence intervals; SLEs = stressful life events; SI = suicidal ideation; T1 = baseline; T2 = 12 weeks; T3 = 24 weeks. ^1^ The structural equation models for SI were adjusted for gender, age, marriage, education level, employed status, family monthly income, smoking, drinking, exercise habit per week, depressive symptom, anxiety symptom, resilience, and SI at baseline; model fit indices (CFI = 1.000; RMSEA = 0.001, SRMR = 0.001).

## Data Availability

The datasets used or analyzed during the current study are available from Prof. Lu on reasonable request.

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
