# Peer review of "Longitudinal Association between Stressful Life Events and Suicidal Ideation in Adults with Major Depression Disorder: The Mediating Effects of Insomnia Symptoms"

_behavsci, 2024, doi:10.3390/bs14060467_

Round 1

Reviewer 1 Report

Comments and Suggestions for Authors

The aim of this article is to evaluate the role of insomnia symptoms in the association between stressful life events and suicidal ideation, using a questionnaire.
The article is well written and clearly organised. The objective of the study is well presented. However, some details could be improved to enhance the reader's understanding and the relevance of the argument.

Author Response

Responses to Reviewer 1 Comments

Dear Reviewer,

Thank you very much for your valuable suggestions, which improved our manuscript. We responded to each comment and made corresponding changes in the manuscript. We believe that the revised manuscript is now much easier for readers to follow. According to these comments and suggestions, we added new information to the revised manuscript to help clarify the content. The response to each comment and corresponding changes are listed as follows. The comments from reviewers are shown in blue, and the responses are shown in black. The revisions in the revised manuscript are shown in red.

Your efforts in reviewing the manuscript are highly appreciated.

Sincerely,

Authors of behavsci-2942485

Reviewer #1:

1. Line 115: in my knowledge, the SLESQ (Goodman et al., 1998) consisted of 13 items. If the authors used another version, they should mention it or they may precise which item was deleted and why.

Revision/Answer: Thank you for your careful review. Given the national context of China, following discussions with pertinent experts, we have decided to exclude the item "Living in environments that are akin to armed conflict or war zones where life-threatening dangers could occur at any moment." This resulted in a 12-item Chinese version of the SLESQ, which was utilized in this study to assess exposure to SLEs. Additionally, we have included the rationale for this exclusion and relevant references to the 12-item version of the scale at the appropriate location in the revised manuscript. (Revised manuscript: the Page 3-4, Line 116-121)

Revised sentence:

The SLESQ consisted of 13 items, which asked participants to report SLEs that occurred in their daily life. Given the national context of China, following discussions with pertinent experts, we excluded the item "Living in environments akin to armed conflict or war zones where life-threatening dangers could occur at any moment." Consequently, this decision led to the development of a 12-item Chinese version of the SLESQ, which was utilized in this study to assess exposure to SLEs [31].

2. Lines 119-122: The authors should precise on what basis (which reference) they considered the 3 categories: 0 / (1-2) / (3 or more), especially as:

- in the descriptive part of the data (lines 187 to 188 and table 1) authors did not take this distinction into account and only considered whether or not participants have had a trauma (regardless of the number).

- in the results section authors considered SLE as a continuous variable.

Revision/Answer: Thank you for your careful review.

In the methodological section, the cumulative number of stressful life event (SLE) experiences was stratified into three levels due to the significant skewness observed in the measurement of SLEs, with the majority (>75%) of participants experiencing between 0 to 3 types of SLEs. The rationale for adopting this stratification method, along with pertinent literature, has been interspersed at the appropriate juncture within the revised manuscript. (Revised manuscript: the Page 4, Line 125-129)

In the descriptive data segment, given that this study is predicated on a cohort study design, the aim was to delineate the disparities between the SLE-exposed group and the control group regarding general demographic characteristics, health-related behaviors, and mental health status. Consequently, SLE exposure was binarized (yes or no) for the purposes of grouping and description.

In the results section, both univariate and multivariate analyses were conducted to explore the relationship between the three-level exposure to SLEs and suicidal ideation among patients with depression. We have revised the sentence in the revised manuscript.

Revised sentence:

Furthermore, due to the significant skewness observed in the measurement of SLEs, with the majority (>75%) of participants experiencing 0 to 3 types of SLEs, we defined 3 levels of SLE exposure: 0 SLE as low exposure, 1 or 2 SLEs as normative exposure and 3 or more SLEs as high exposure. This classification aimed to ensure a relatively even distribution across groups [32].

3. Line 127-129: It is not clear to me; did the authors only use 5 of the 19 items of the BSSI? If yes, they may justify their choice.

Revision/Answer: Thank you for your careful review. The decision to assess suicidal ideation (SI) in patients with MDD using only the first 5 items of the BSSI is based on two primary reasons.

Firstly, according to the scale's instructions, the intensity of SI is determined by the first 5 items. The risk of suicide is assessed based on items 6 to 19, which determine the likelihood of a participant with SI actually attempting suicide. Thus, the first 5 items may more accurately reflect the intensity of SI among the study subjects.

Secondly, items 6–19 are only completed if the respondent rates items 4 or 5 with a score of 1 or greater, and thus we had more complete data for part 1 score. Additionally, the use of the first 5 items to assess SI in the study subjects is supported by previous research (Hawes, Mariah et al. “Anhedonia and suicidal thoughts and behaviors in psychiatric outpatients: The role of acuity.” Depression and anxiety vol. 35,12 (2018): 1218-1227. doi:10.1002/da.22814).

To elucidate our choice of using only the first 5 items of the scale to assess SI in the study subjects, we have supplemented the corresponding explanations in the revised manuscript. (Revised manuscript: the Page 4, Line 136-140)

Revised sentence:

The intensity of SI is assessed by the first 5 items. The risk of suicide is evaluated based on items 6 to 19, which determine the likelihood of a participant with SI actually attempting suicide. Items 6–19 are only completed if the respondent rates items 4 or 5 with a score of 1 or greater, and thus we had more complete data for the first 5 items. Consequently, in this study, SI was assessed using the first 5 items, with higher scores indicating higher levels of SI.

4. Line 152: A capital letter is missing

Revision/Answer: Thank you for your careful review. We have replaced the lowercase 'h' with an uppercase 'H'. Thank you very much for your careful review, and we apologize for such an error. We have thoroughly checked the entire document to ensure there are no similar minor mistakes. We have revised the sentence in the revised manuscript. (Revised manuscript: the Page 4, Line 165)

Revised sentence:

Have you ever consumed at least one alcoholic drink of any kind?

5. Lines 152-154: How did participants respond to these questions? By “Yes” or “No”?

Revision/Answer: Thank you for your careful review. Actually, participants responded to these questions with “Yes” or “No”. We have revised the sentence in the revised manuscript. (Revised manuscript: the Page 4, Line 165-167)

Revised sentence:

Lifetime smoking, life drinking and weekly exercise habit were evaluated by the following questions: “Have you ever smoked a cigarette (0 = No; 1 = Yes)? Have you ever consumed at least one alcoholic drink of any kind (0 = No; 1 = Yes)? Do you have a weekly exercise for more than 30 minutes (0 = No; 1 = Yes)?”.

6. In the results section:

Lines 197-200: all the results of the Chi-square tests should be reported (at least in the supplement).

Revision/Answer: Thank you for your careful review. An additional column has been incorporated into Table 1 to display the outcomes of the chi-square test. (Revised manuscript: the Page 6, Line 219-220)

Table 1. Baseline characteristics of 511 participants

Factor

Total,

n (%)

Stressful life events

ê­“2

P value

Yes

No

Total

511(100.0)

342(66.9)

169(33.1)

NA

NA

Gender (N, %)

Male

168(32.9)

117(34.2)

51(30.2)

0.834

0.361

Female

343(67.1)

225(65.8)

118(69.8)

Age (mean, SD)

28.7(6.7)

28.6(6.1)

28.8(7.0)

0.228

0.820

Marriage (N, %)

Unmarried

379(74.2)

261(76.3)

118(69.8)

2.808

0.094

Married

122(23.9)

74(21.6)

48(28.4)

Missing

10(2.0)

7(2.0)

3(1.8)

Education level (N, %)

Below undergraduate

220(43.1)

155(45.3)

65(38.5)

2.171

0.141

Undergraduate or above

291(56.9)

187(54.7)

104(61.5)

Employed status (N, %)

Unemployed

158(30.9)

109(31.9)

49(29.0)

0.466

0.495

Employed

352(68.9)

232(67.8)

120(71.0)

Missing

1(0.2)

1(0.3)

0(0.0)

Family monthly income (N, %)

<10,000¥

187(36.6)

132(38.6)

55(32.5)

0.756

0.385

≥10,000¥

286(56.0)

191(55.8)

95(56.2)

Missing

38(7.4)

19(5.6)

19(11.2)

Smoking (N, %)

No

301(58.9)

189(55.3)

112(66.3)

5.169

0.023

Yes

207(40.5)

150(43.9)

57(33.7)

Missing

3(0.6)

3(0.9)

0(0.0)

Drinking (N, %)

No

79(15.5)

49(14.3)

30(17.8)

0.987

0.320

Yes

431(84.3)

292(85.4)

139(82.2)

Missing

1(0.2)

1(0.3)

0(0.0)

Note. 95% CI=95% confidence intervals; SI= Suicidal ideation; ISI=Insomnia Severity Index.

Revised sentence:

Table 1. Baseline characteristics of 511 participants (cont.)

Factor

Total,

n (%)

Stressful life events

ê­“2

P value

Yes

No

Exercise habit per week (N, %)

No

329(64.4)

224(65.5)

105(62.1)

0.625

0.429

Yes

181(35.4)

117(34.2)

64(37.9)

Missing

1(0.2)

1(0.3)

0(0.0)

Moderate or severe depressive symptom (N, %)

No

29(5.7)

14(4.1)

15(8.9)

4.832

0.028

Yes

482(94.3)

328(95.9)

154(91.1)

Moderate or severe anxiety symptom (N, %)

No

113(22.1)

72(21.1)

41(24.3)

0.648

0.421

Yes

397(77.7)

269(78.7)

128(75.7)

Missing

1(0.2)

1(0.3)

0(0.0)

Resilience at baseline (N, %)

Weak resilience (≤60)

471(92.2)

317(92.7)

154(91.1)

0.097

0.755

Greater resilience

34(6.7)

22(6.4)

12(7.1)

Missing

6(1.2)

3(0.9)

3(1.8)

SI at baseline (mean, SD)

3.8(2.6)

4.0(2.7)

3.4(2.5)

-2.703

0.007

ISI scores (mean, SD)

14.9(6.3)

15.0(6.6)

14.8(6.2)

-0.278

0.781

Note. 95% CI=95% confidence intervals; SI= Suicidal ideation; ISI=Insomnia Severity Index.

7. In the Discussion section:

Line 249-250: The authors refer to the prevalence of insomnia in their population, but if I'm not mistaken, this result is not presented anywhere before.

Table S1. Baseline sample characteristics between eligible and ineligible participants. (cont.)

Factor

Total, n (%)

Eligible

Ineligible

P value

SI at baseline (N, %)

No

224(20.4)

105(20.5)

119(20.3)

0.990

Yes

871(79.4)

405(79.3)

466(79.5)

Note. SLEs= Stressful life events; SI= Suicidal ideation; ISI=Insomnia Severity Index.

Revision/Answer: Thank you for your careful review. In fact, since SI is an abbreviation for suicidal ideation, the prevalence we discuss pertains to this condition. The prevalence of suicidal ideation is indeed not depicted in the main text's table but is illustrated in Supplementary Materials, specifically in Table S1. The source of the data has been clearly indicated within the manuscript. The sentence has been revised accordingly. (Revised manuscript: the Page 10, Line 264-265)

Revised sentence:

In addition, as illustrated in Table S1, SI was also common among persons with MDD, with the prevalence of 79.3% in our study.

Reviewer 2 Report

Comments and Suggestions for Authors

This is a well conducted interesting longitudinal study about the role of stressful life events and insomnia in suicidal ideation of people with major depressive disorder (MDD). Overall, results provide a better understanding of MDD population and are of great value for clinical interventions.

I just have some minor comments:

Introduction

-              In page 2 you state “Stressful life events (SLEs) are common stressors for suicide”. This is a really broad sentence since stressful live events can be every stressor. Please be more concrete about which kind of stressful life events can trigger suicidal ideation. 

Methods

-              Can you provide information about SLE questionnaire Cronbach’s alpha for this study?

Author Response

Responses to Reviewer 2 Comments

Dear Reviewer,

Thank you very much for your valuable suggestions, which improved our manuscript. We responded to each comment and made corresponding changes in the manuscript. We believe that the revised manuscript is now much easier for readers to follow. According to these comments and suggestions, we added new information to the revised manuscript to help clarify the content. The response to each comment and corresponding changes are listed as follows. The comments from reviewers are shown in blue, and the responses are shown in black. The revisions in the revised manuscript are shown in red.

Your efforts in reviewing the manuscript are highly appreciated.

Sincerely,

Authors of behavsci-2942485

Reviewer #2:

1. Introduction

In page 2 you state “Stressful life events (SLEs) are common stressors for suicide”. This is a really broad sentence since stressful live events can be every stressor. Please be more concrete about which kind of stressful life events can trigger suicidal ideation.

Revision/Answer: Thank you for your careful review. In accordance with your suggestions, I have provided a supplementary explanation of the definition of Stressful life events (SLEs). Furthermore, in the subsequent lines 59 to 61, the text elucidates which specific types of stressors are implicated in the emergence of suicidal ideation (Second, prior studies have investigated risk factors associated with suicidal behaviors in different populations, including specific stressors such as childhood trauma [13–15], interpersonal stress [16,17], bullying [18], violence [19] and so on.). We have revised the sentence in the revised manuscript. (Revised manuscript: the Page 2, Line 49-51)

Revised sentence:

SLEs refer to significant incidents that occur abruptly and provoke strong psychological responses in individuals.

2. Methods

Can you provide information about SLE questionnaire Cronbach’s alpha for this study?

Revision/Answer: Thank you for your careful review. After calculation, the Cronbach's alpha coefficient for the SLE questionnaire used in this study is 0.70, indicating acceptable internal consistency. We have revised the sentence in the revised manuscript. (Revised manuscript: the Page 4, Line 129-130)

Revised sentence:

Additionally, the SLESQ was utilized at baseline and shown acceptable internal consistency in the current study (Cronbach's α =0.70).